# Research on a Small-Sample Fault Diagnosis Method for UAV Engines Based on an MSSST and ACS-BPNN Optimized Deep Convolutional Network

Siyu Li [1,†], Zichang Liu [1,†], Yunbin Yan [1], Kai Han [1], Yueming Han [1], Xinyu Miao [2], Zhonghua Cheng [1,*] and Shifei Ma [3]

1    Shijiazhuang Campus, Army Engineering University of PLA, Shijiazhuang 050003, China; sy_li19880806@163.com (S.L.); zc_liu1997@aeu.edu.cn (Z.L.); 17636697399@163.com (Y.Y.); haihang67151163@163.com (K.H.); hanyueming2023@163.com (Y.H.)
2    Armed Police Beijing Municipal Command Sixth Detachment on Duty, Beijing 100073, China; 15510711666@163.com
3    Shijiazhuang Division of PLAA Infantry Academy, Shijiazhuang 050003, China; 18031292999@163.com
*    Correspondence: a15032073178@sina.com; Tel.: +86-0311-87994538
†    These authors contributed equally to this work.

**Abstract:** Regarding the difficulty of extracting fault information in the faulty status of UAV (unmanned aerial vehicle) engines and the high time cost and large data requirement of the existing deep learning fault diagnosis algorithms with many training parameters, in this paper, a small-sample transfer learning fault diagnosis algorithm is proposed. First, vibration signals under the engine fault status are converted into a two-dimensional time-frequency map by multiple simultaneous squeezing S-transform (MSSST), which reduces the randomness of manually extracted features. Second, to address the problems of slow network model training and large data sample requirement, a transfer diagnosis strategy using the fine-tuned time-frequency map samples as the pre-training model of the ResNet-18 convolutional neural network is proposed. In addition, in order to improve the training effect of the network model, an agent model is introduced to optimize the hyperparameter network autonomously. Finally, experiments show that the algorithm proposed in this paper can obtain high classification accuracy in fault diagnosis of UAV engines compared to other commonly used methods, with a classification accuracy of faults as high as 97.1751%; in addition, we show that it maintains a very stable small-sample migratory learning capability under this condition.

**Keywords:** fault diagnosis; transfer learning; surrogate model; hyperparameter optimization; small sample





## 1. Introduction

UAVs have a wide range of uses, with the advantages of low cost, high efficiency, strong survivability, and good mobility. They can carry out rescue and disaster relief, agricultural irrigation, power patrol, be used in environmental protection, film shooting, border patrol, military operations, and work in other fields [1,2]. The engine is the component with the highest failure rate, the most complex performance correction, and the largest maintenance workload in the UAV system. Its working status directly affects the safe and reliable flight of the UAV. Due to the complex structure and function of UAV engines and the harsh working environment, the requirements for reliability, stability, and bearing capacity are very high [3]. However, the long-time operation of the engine brings a series of problems, such as wear of mechanical parts, load failure, corrosion, defects, etc. These damage conditions cause an increase in the fuel consumption rate, exhaust temperature, a decrease of thrust, etc., and gradually aggravates performance degradation of engine parts and the whole machine, leading to the occurrence of faults. In serious cases, they cause

complete failure of engine performance or even shutdown [4]. Therefore, fault detection and diagnosis of the working status of UAV engines are of great significance.

Traditional fault diagnosis comprises three steps: signal processing, feature extraction, and fault status recognition [5]. Tang designed a four-channel sound signal acquisition system for UAV engine testing and performed blind source separation on the collected signals. First, the correlation coefficient is used as the standard to collect four signals, and then three of them are selected for underdetermined blind source separation using a blind source separation algorithm. Finally, four kinds of signals are obtained, namely engine propeller noise, exhaust noise, fan noise in the test room, and other background noise, which provide necessary data for further engine fault location and diagnosis [6]. Regarding the problem of low accuracy of UAV sensor fault diagnosis, Ye et al. decomposed the collected UAV sensor original signal through four layers of wavelet decomposition and classified the fault status as the input of a multi-core support vector machine. The classification accuracy reaches 91%, at which the fault diagnosis of UAV sensors can be carried out effectively [7]. Traditional fault diagnosis methods need to process the original signal on a large scale and cannot effectively reflect the logical relationship between the processed signal and the fault, which restricts the improvement of the accuracy of fault diagnosis [8].

The fault diagnosis method based on deep learning can adaptively extract features according to the characteristics of the data themselves, without manual feature extraction, and can directly take time series data and high-dimensional data as input and enhance the nonlinear processing ability of the model by increasing the number of layers of the neural network [9]. At the same time, the deep learning method has strong generalization ability. With the support of large-scale data sets, the model can be better generalized to unknown fault modes and has certain adaptability to new fault diagnosis problems [10]. In reference [11], regarding the problem whereby the UAV system cannot accurately carry out fault monitoring and diagnosis, the collected vibration signal was input into the convolutional neural network with three layers of convolution layer and pooling layer, and the accuracy of fault diagnosis reaches 97.5%. Ma [12] designed a UAV experimental acquisition system based on the vibration signal, which is used for nondestructive UAV testing and fault diagnosis methods. Four kinds of vibration signals were collected in the experiment, namely normal status and three kinds of fault status. First, empirical mode decomposition (EMD) was used to reduce the noise of the original signal. Second, the *y*-axis data were screened out by the root mean square error (RMSE). Finally, it was input into the convolutional neural network for fault diagnosis, which was verified. It had a certain reference value for nondestructive testing and fault diagnosis of multi-rotor UAV. In the above research, the deep learning model was trained with one-dimensional signals, but the data information contained in two-dimensional images was more comprehensive [13]. Its main advantages are as follows:

1. Two-dimensional images can represent data information from multiple angles, and the information represented by one-dimensional signals is not comprehensive.
2. The image is easier to distinguish. Through the time-frequency conversion method, the one-dimensional signal is converted into a two-dimensional time-frequency map. Through the intelligent classification method, the classification and recognition can be more intuitive.

Through the summary of the above literature, it can be found that the current research on UAV engine fault diagnosis still has the following problems: first, the traditional time-frequency analysis methods, such as continuous wavelet transform, Hilbert–Huang transform, and short-time Fourier transform, still have room for improvement in time-frequency resolution, which has an important impact on the accuracy of fault diagnosis based on deep learning; second, the research based on UAV engines is lacking, and the amount of data that can be used as training samples is not rich enough. Therefore, in order to solve the above problems, this paper proposes a small-sample migration learning fault

diagnosis method for the UAV engine based on MSSST and ACS-BPNN surrogate model optimization depth convolution, which can accurately carry out fault diagnosis.

The main contributions of this study are as follows:

(a)  Using the current experimental environment, the preset fault experiment of the UAV engine is carried out, and the vibration signal of the UAV engine in a typical fault status is collected.

(b)  The one-dimensional vibration signal of the UAV engine is transformed into a time-frequency map by MSSST, and the time dependence of the vibration signal is mapped into the image feature space, so that the original feature information is retained in the time-frequency map as much as possible. Then, the deep learning network trained by small-sample transfer learning is used to automatically extract the temporal and spatial features in the image and complete the fault status recognition of small samples.

(c)  The feasibility and effectiveness of the proposed fault diagnosis method for the UAV engine are verified by the measured data of the UAV engine.

The rest is as follows: Section 2 introduces the relevant theories of MSSST and small-sample transfer learning in detail; Section 3 introduces the process of small-sample transfer learning fault diagnosis; in Section 4, the preset fault experiment of the UAV engine and the process of vibration data acquisition are introduced, and the experimental results are analyzed and studied; finally, the experimental results and the future research prospects of UAV engine fault diagnosis are put forward.

## 2. Relevant Theories

### 2.1. Time-Frequency Image Conversion Based on MSSST

In the fault diagnosis process, the acquisition devices usually collect 1D signals. Both evolutionary networks in this paper use images as inputs, so it is necessary to convert the 1D vibration signals into 2D color time-frequency maps. One conversion method is to directly intercept the vibration signal at equal intervals and reorganize it into a two-dimensional matrix, which is then saved as a time-frequency map. However, this method does not reflect the frequency domain characteristics of the signal. Based on the time-frequency domain transform methods, there are mainly the Short-Time Fourier Transform (STFT), the Continuous Wavelet Transform (CWT) and the Hilbert–Huang transform [14].

Linear time-frequency analysis methods such as STFT, CWT and Hilbert–Huang transform have the advantages of simple computation and perfect reconstruction formulas [15–17]. At the same time, it is clear from linear transformation that no cross terms are generated in the time-frequency domain. However, there are two common limitations common to the above methods. First, the time-frequency resolution is limited; second, under the uncertainty principle, the time-frequency analysis using the above methods cannot localize the signal in time and frequency with high resolution.

#### 2.1.1. Synchronous Compression S-Transform

The S-transform inherits and develops the advantages of STFT and CWT, but the Gaussian window function used in the S-transform is still a fixed window, which cannot be adjusted in real time, so the S-transform as a means of time-frequency analysis still has room for improvement [18]. There are two ways to enhance the analytical ability of S-transform time-frequency: one is by adjusting the window function so that the window function can be adapted to the characteristics of the signal in order to obtain the time-frequency image with higher time-frequency resolution, but the selection of the window function is a difficult problem at present, and it is not easy to be realized; the second way to enhance the time-frequency analysis of the S-transform is more commonly used, and it is a combination of the synchronous compression idea and the time-frequency analysis method, i.e., compression of S-transform coefficients within a certain interval to a point, so that the time-frequency image with a high time-frequency resolution can be obtained. According to the principle of wavelet synchronous compression transform, the process of Synchro squeezing S-transform (SSST) can be derived [19].

The formula for the S-transform can be expanded and written in the following form:

$$ST(\tau,f) = \frac{|f|}{\sqrt{2\pi}} \int_{-\infty}^{+\infty} x(t)e^{\frac{-(t-\tau)^2 f^2}{2}} e^{-i2\pi f(t-\tau)} e^{-i2\pi f\tau} dt. \tag{1}$$

We make $\psi(t) = \frac{1}{\sqrt{2\pi}} e^{-\frac{t^2}{2}} e^{i2\pi t}$; then, Equation (1) can be expressed as

$$ST(\tau,f) = |f|e^{-i2\pi f\tau} \int_{-\infty}^{+\infty} x(t)\overline{\psi[f(t-\tau)]}\, dt, \tag{2}$$

where $\psi(t)$ is a complex conjugate of function $\overline{\psi(t)}$. According to the Parseval theorem and the Plancherel theorem, Equation (2) can be written as

$$ST(\tau,f) = |f|e^{-i2\pi f\tau} \int_{-\infty}^{+\infty} \hat{x}(\xi)\overline{\hat{\psi}[f^{-1}\xi]} e^{i\tau\xi} d\xi, \tag{3}$$

where $\hat{x}(\xi)$ is the Fourier transform of signal $x(t)$. $\hat{\psi}(\xi)$ is the complex conjugate of Fourier transform of $\psi(t)$.

When signal $x(t)$ is a harmonic component, i.e., $x(t) = A\cos 2\pi f_0 t$,

$$\hat{x}(\xi) = A\pi[\delta(\xi - 2\pi f_0) + \delta(\xi + 2\pi f_0)]. \tag{4}$$

Substituting Equation (4) into Equation (3), the following equation can be obtained:

$$ST(\tau,f) = \frac{A}{2}e^{-i2\pi(f-f_0)\tau} \overline{\hat{\psi}(2\pi f^{-1} f_0)}. \tag{5}$$

We calculate the partial derivative of Equation (5) to obtain

$$\frac{ST(\tau,f)}{\partial t} = iA\pi(f_0 - f) e^{-i2\pi(f-f_0)\tau} \overline{\hat{\psi}(2\pi f^{-1} f_0)}. \tag{6}$$

Therefore, the instantaneous frequency of signal $x(t)$ is

$$f(\tau,f) = f + [i2\pi ST(\tau,f)]^{-1} \frac{\partial ST(\tau,f)}{\partial \tau}. \tag{7}$$

Obviously, for the single component signal of $x(t) = A\cos 2\pi f_0 t$, from Equation (7), the following can be obtained:

$$\begin{aligned}
f(\tau,f) &= f + [i2\pi ST(\tau,f)]^{-1} \frac{\partial ST(\tau,f)}{\partial \tau} \\
&= f + \frac{iA\pi(f_0-f) e^{-i2\pi(f-f_0)\tau} \hat{\psi}(2\pi f^{-1} f_0)}{i\pi \frac{A}{2} e^{-i2\pi(f-f_0)\tau} \hat{\psi}(2\pi f^{-1} f_0)} \\
&= f_0
\end{aligned} \tag{8}$$

For more general multicomponent signals,

$$x(t) = \sum_{n=1}^{N} x_n(t) = \sum_{n=1}^{N} A_n(t)\cos[\phi_n(t)]. \tag{9}$$

It satisfies that for any time $t$ obtained, $A_n(t)$ will have $\phi\prime_n(t) > 0$, which is the derivative of $\phi_n(t)$.

When $x(t)$ is a multicomponent signal, considering the linear property of the S-transform, the result of its S-transform can be expressed as a superposition of the S-transforms of $N$ components, $x_n(t)$, with expression

$$ST_x(\tau,f) = \sum_{n=1}^{N} ST_{x_n}(\tau,f_n), \tag{10}$$

$$ST_{x_n}(\tau, f_n) = \frac{|f_n|}{\sqrt{2\pi}} \int_{-\infty}^{+\infty} x_n(t) e^{-\frac{(\tau-t)^2 f_n^2}{2}} e^{-i2\pi f_n t} dt. \tag{11}$$

The instantaneous frequency of signal component $x_n(t)$ can be expressed as

$$f_{x_n}(\tau, f_n) = f_n + [i2\pi ST_{x_n}(\tau, f_n)]^{-1} \frac{\partial ST_{x_n}(\tau, f_n)}{\partial \tau}. \tag{12}$$

Then, the instantaneous frequency of signal $x(t)$ can be expressed in the form of the summation of component signals:

$$f_x(\tau, f) = \sum_{n=1}^{N} \left\{ \delta(f - f_n) \left[ f_n + (i2\pi ST_{x_n}(\tau, f_n))^{-1} \frac{\partial ST_{x_n}(\tau, f_n)}{\partial \tau} \right] \right\}, \tag{13}$$

where $\delta$ is the impulse function.

The basic principle of SSST is compressing the value in surrounding section $\left[ f_l - \frac{1}{2}\Delta f_l, f_l + \frac{1}{2}\Delta f_l \right]$ of center instantaneous frequency $f_l$ to $f_l$. The integral expression of the sum of SSST on a continuous interval is

$$SSST_x(\tau, f_l) = \frac{1}{2\Delta f_l} \int_{f_l - \frac{1}{2}\Delta f_l}^{f_l + \frac{1}{2}\Delta f_l} |ST_x(\tau, f)| f_x(\tau, f) df. \tag{14}$$

Similarly, when the continuous integral is in the form of discrete summation, the expression of SSST under discrete conditions can be obtained as follows:

$$SSST_x(\tau, f_l) = \frac{1}{2\Delta f_l} \sum_{f_k : |f_x(\tau, f_k) - f_l| \leq \Delta f_k / 2} |ST_x(\tau, f_k)| f_k \Delta f_k, \tag{15}$$

where $f_l$ is frequency after SSST; $f_k$ is discrete frequency samples of the frequency interval on the S-transform spectrum, and $\Delta f_k = f_k - f_{k-1}$.

Taking the absolute amplitude of the S-transform result in Equations (14) and (15) avoids the loss of energy due to the positive and negative summation in the compression process, which can effectively improve the compression effect of the S-transform and improve the time-frequency energy aggregation of the signal.

2.1.2. Multiple Simultaneous Squeezing S-Transform

The resulting time-frequency spectrograms have a high resolution when SSST is used to process weak time-varying signals. However, for high-frequency weak-amplitude signals, the time-frequency spectrograms processed by SSST still have the problem of energy leakage. For this reason, this paper combines the simultaneous compressed S-transform with the idea of multiple iterations to propose a new method of time-frequency analysis based on Multi-SSST(MSSST), which is used to gradually improve the time-frequency resolution of time-frequency spectrograms through multiple iterations sequentially in order to reduce energy leakage.

The iterative formula for MSSST is as follows:

$$
\begin{aligned}
Ts^{[1]} &= SSST_x(\tau, f_l) \\
Ts^{[2]}(\tau, f_l) &= \int_{f_l - \frac{1}{2}\Delta f_l}^{f_l + \frac{1}{2}\Delta f_l} Ts^{[1]}(\tau, \xi) f[\tau, f(\tau, \xi)] \mathrm{d}\xi \\
&= \int_{f_l - \frac{1}{2}\Delta f_l}^{f_l + \frac{1}{2}\Delta f_l} ST_x(\tau, f) f_x(\tau, f) f[\tau, f(\tau, \xi)] \mathrm{d}\xi \\
Ts^{[3]}(\tau, f_l) &= \int_{f_l - \frac{1}{2}\Delta f_l}^{f_l + \frac{1}{2}\Delta f_l} Ts^{[2]}(\tau, \xi) f[\tau, f(\tau, \xi)] \mathrm{d}\xi \\
&= \int_{f_l - \frac{1}{2}\Delta f_l}^{f_l + \frac{1}{2}\Delta f_l} ST_x(\tau, f) f_x(\tau, f) f[\tau, f(\tau, \xi)] f[\tau, f(\tau, \xi)] \mathrm{d}\xi \\
&\vdots \\
Ts^{[N]}(\tau, f_l) &= \int_{f_l - \frac{1}{2}\Delta f_l}^{f_l + \frac{1}{2}\Delta f_l} Ts^{[N-1]}(\tau, \xi) f(\tau, f(\tau, \xi)) \mathrm{d}\xi \\
&= \int_{f_l - \frac{1}{2}\Delta f_l}^{f_l + \frac{1}{2}\Delta f_l} ST_x(\tau, f) f_x(\tau, f) f[\tau, f(\tau, \xi)] \cdots f[\tau, f(\tau, \xi)] \mathrm{d}\xi
\end{aligned} \tag{16}
$$

where $N$ denotes the number of iterations ($N \geq 2$). Taking $Ts^{[k-1]}$ into $Ts^{[k]}$ ($k = 2, 3, \ldots, N$) and after $N - 1$ calculations, final result $Ts^{[N]}(\tau, f_l)$ can be reduced to

$$
\begin{aligned}
MSSST_x(\tau, f_l) &= Ts^{[N]}(\tau, f_l) \\
&= \int_{f_l - \frac{1}{2}\Delta f_l}^{f_l + \frac{1}{2}\Delta f_l} ST_x(\tau, \xi) f_x(\tau, \xi) f[\tau, f(\tau, \xi)]^{[N-1]} \mathrm{d}\xi
\end{aligned} \tag{17}
$$

In practice, performing SSST sequentially creates a huge computational burden. Equation (17) is a simplified formula after many iterations, and when calculating the MSSST of order $N$ according to Equation (17), it is only necessary to calculate $f[\tau, f(\tau, \xi)]^{[N-1]}$ first, and then bring it into Equation (13) to obtain the same calculation result as that of Equation (16), and the computational complexity of Equation (17) is not different from that of the first-order SSST, which is conducive to improving calculation speed.

Compared with the STFT, the MSSST can greatly improve the time-frequency resolution of the STFT, which is conducive to further improving the discriminative power of network model fault diagnosis.

### 2.2. Transfer Learning

Transfer learning is a new deep learning approach the goal of which is to extract similar components (transfer components) between different but related domains so that knowledge learned on one domain can be applied to another domain [20]. The original data domain is called the source domain, and the target domain of interest is called the target domain. To facilitate description, the concepts of domain and task are introduced. Domain $\Omega$ usually consists of a collection of feature vectors and a data distribution, $X$ is defined as a collection containing $n$ samples, and the corresponding feature vectors of each sample together form a collection of feature vectors (feature space), which in turn can be used to define a domain by $\Omega = \{\chi, P(X)\}$, where $\chi$ is the feature space, $P(X)$ is the marginal probability distribution of the sample, and $X = \{x_1, x_2, \ldots, x_n\} \subset \chi$. Further, the concept of a task is introduced, i.e., for an input feature vector, a reasonable prediction of the vector's label is made probabilistically by means of label prediction function $f(\cdot)$. A task can thus be defined in terms of $\gamma = \{\varphi, P(Y|X)\}$, where $\varphi$ is the labeling space and $P(Y|X)$ is the labeling category of the vector. Given the concepts of domain and task, transfer learning is mainly used to solve the problem of knowledge learning and transfer under different domain and different task conditions (see Figure 1) [21].

In terms of technical tools, transfer learning can be categorized into instance-based transfer learning, feature-based transfer learning, association-based transfer learning and parameter-based transfer learning [22]. Instance-based transfer learning improves the effectiveness and robustness of transfer learning by adjusting the weights of the parts of the source domain that are more similar to the target domain. Feature-based transfer

learning attempts to construct a feature subspace that integrates the shared feature factors of the source and target domains, which can reduce the feature differences between the two and improve the transferability of knowledge. The purpose of association rule-based transfer learning is to discover potential connections between source and target domains, focusing on transferability. Parameter-based transfer learning reduces the differences between domains with the idea of using a large amount of source-domain data to train the model under the transfer learning strategy of shallow parameter freezing and deep parameter learning, and then use a small amount of target-domain data to determine the depth parameters of the model so that the parameters of the deep network layer are more in line with the classification features of the target-domain data.

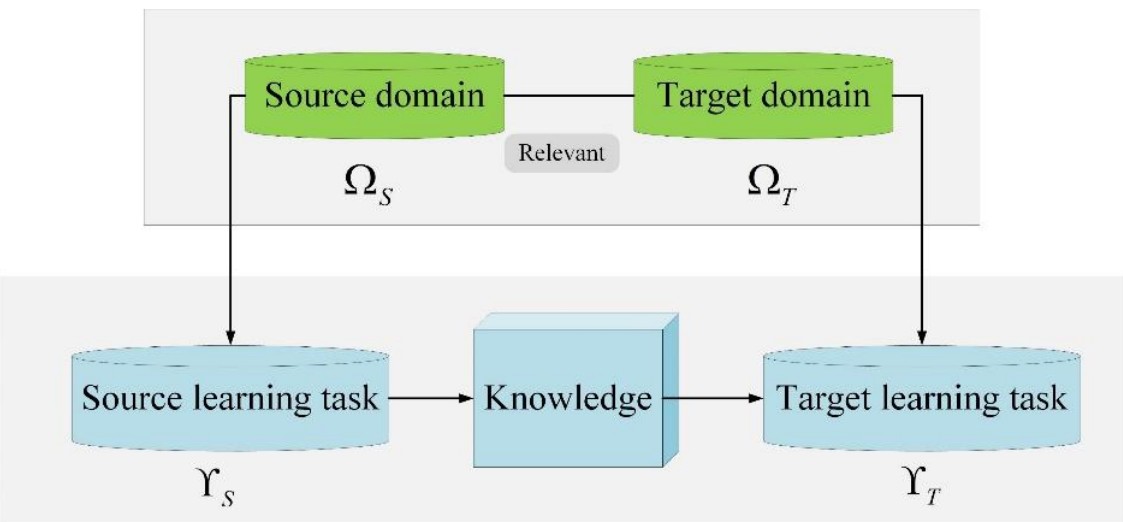

**Figure 1.** Schematic diagram of transfer learning.

The source domain knowledge is a sufficient number of ImageNet datasets, the target domain is time-frequency image samples of UAV engine vibration signals, the network models are AlexNet Convolutional Networks and ResNet-18 Convolutional Networks, and the migration strategy uses the freezing of the shallow network layers and fine-tuning of the deeper network layers. This method eliminates the need for end-to-end training of the network model as well as the computation of the difference metric between the source and target domain data at each iteration and the need for reverse iteration. Only a small number of samples are needed to fine-tune the depth classification parameters of the network. In this way, the classification layer has the characteristics of edge distribution of the target domain data. The network's ability of depth feature extraction is utilized for pictures in order to determine the subtle differences between the time-frequency images of the different faults in the engine of the unmanned aerial vehicle so as to achieve the purpose of rapid classification of faults under the condition of a small number of samples of equipment.

### 2.3. Convolutional Network Model

In recent years, deep learning networks have been enriched and developed, among which Alexnet, Googlenet and ResNet are typical high-quality convolutional network models and are widely used. These models already have some parameter bases after learning from the ImageNet image set. For data processing in other domains, it is necessary to utilize migration learning, but for the processing of small-sample data in this paper, too many network layers and complex structural models tend to have problems with training speed and overfitting of training results. Therefore, in this paper, the ResNet-18 convolutional network with simple network structure is chosen as the deep learning

processing algorithm, and a migration learning strategy is used to realize the migration detection between the ImageNet image set and the data in this paper [23].

The ResNet-18 network is a typical deep residual network, and the introduction of residual mapping improves the learning ability of the network, thus speeding up the convergence of the model. The structural model in ResNet-18 mainly consists of a convolutional layer, four residual layers (basic layer), an average pooling layer and a fully connected layer, where the individual residual blocks are connected to each other by two convolutional layer jumps. Distinguishing this model from the AlexNet model, which acts as a directed acyclic network, the global average pooling layer needs to be connected to the fully connected layer in order to ensure proper network delivery after changes are made to the network layers at the end. The network structure of the ResNet-18 model is illustrated in Figure 2. Only the convolutional and fully connected layers with parameter space are shown in the figure, while the pooling and activation layers are omitted here.

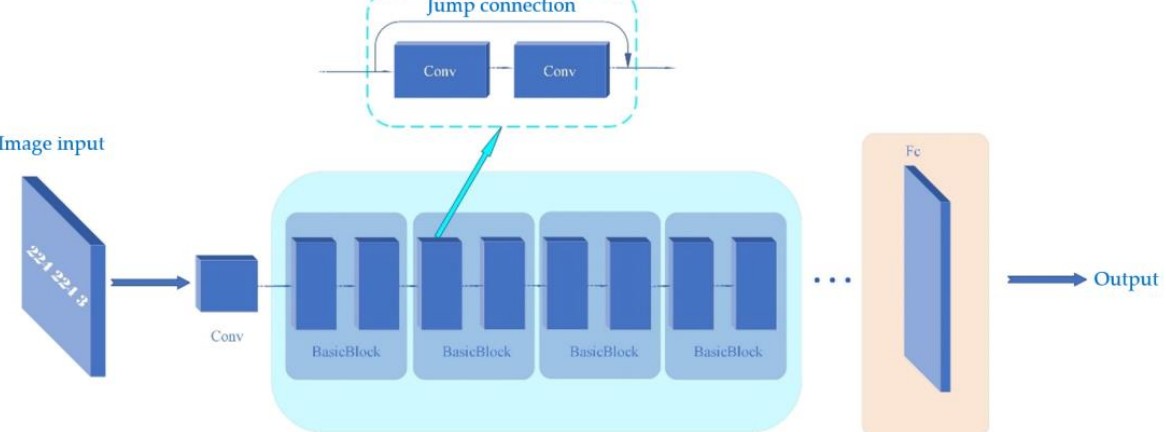

**Figure 2.** Network architecture diagram of ResNet-18.

As can be seen in Figure 2, the two convolutional models have different size requirements for the input of the feature graph, where the Alexnet model requires the input size of the image to be $227 \times 227 \times 3$, while the ResNet-18 model requires the input size of the image to be $224 \times 224 \times 3$. This implies that the sparse representation of the two-dimensional image has to be graphically resized prior to the input to cope with the demands of the different convolutional models.

### 2.4. Agent Model

In the optimization of network hyper-parameters, three key hyper-parameters are optimized, i.e., initial learning rate, batch size and maximum number of trainings. For supervised learning, an appropriate initial learning rate allows for the objective function to converge to a local minimum within the validation time; an appropriate batch size improves the accuracy of gradient descent, thus reducing the magnitude of fluctuations during training; the maximum number of training times determines the degree of convergence of the network, and an excessively small number of training times leads to early convergence of the network, while an excessively large number of training times wastes time [24].

The agent model consists of two main parts: the swarm optimization algorithm and the calculation of the fitness function value. The swarm optimization algorithm adopts an improved kind of Cuckoo Search (CS) population optimization algorithm, which improves the convergence speed and global optimization ability of the Cuckoo Optimization algorithm and helps to optimize the hyperparameters of the network model effectively. For solving fitness function values, the network is trained using different hyperparameter samples to obtain the corresponding training accuracy. The trained BPNN model is embed-

ded in the solution of the fitness function for effective and automatic optimization of the hyperparameters of the evolutionary network.

### 2.4.1. Cuckoo Algorithm and Its Improvement

In nature, the way cuckoos search for suitable locations for their egg-laying nests is actually an optimization process. In view of this, Yang et al. at Cambridge University proposed a population optimization algorithm for CS based on the principle of bionics. The algorithm includes two core ideas: nest parasitism of cuckoos and the Levy (Levy) flight mechanism [25].

Cuckoo breeding is parasitic behavior. Cuckoos themselves do not build nests; when breeding, cuckoos lay their eggs in other hosts' nests and remove some of the host's eggs to improve the survival chances of the parasitic eggs, but the parasitic eggs are sometimes found and discarded by the hosts. In the correspondence between the virtual cuckoo breeding strategy and the solution set, the algorithm considers the whole search space as the feasible domain, the parasitized cuckoo eggs as the problem solution, and the best adapted parasitized cuckoo eggs as the optimal solution of the problem. To better model the cuckoo derivation strategy, the algorithm presupposes three rules:

(a) Cuckoos lay one egg at a time and parasitize a random host nest.
(b) Only the finest eggs are kept for the next generation.
(c) The number of nests is fixed, and $P_a$ is the probability that a parasitized bird's egg is found. Once a parasitized bird egg is found and discarded, the cuckoo then flies throughout the search space, Lévy generating a new nest-finding path, and re-lays the egg.

With the host nest as the search space and cuckoo eggs as the solution, the iteration process from generation $t$ to generation $t + 1$ in the $i$th nest can be expressed as follows:

$$x_i^{(t+1)} = x_i^{(t)} + R\frac{u}{|v|^{1/\beta}}\left[x_i^{(t)} - x_{\text{best}}^{(t)}\right], \tag{18}$$

where $x_i^{(t)}$ and $x_i^{(t+1)}$ are the solutions of generation $t$ and generation $t + 1$ in the $i$ bird's nest, respectively; $R$ is the step size, $R = 0.01$; $u$ and $v$ both obey the normal distribution:

$$\sigma_u = \left\{\frac{\Gamma(1+\beta)\sin(\pi\beta/2)}{\Gamma[(1+\beta)/2]\beta2^{(\beta-1)/2}}\right\}^{1/\beta}, \tag{19}$$

where $\Gamma(z)$ is the gamma function, $\Gamma(z) = \int_0^\infty t^{z-1}\mathrm{e}^{-t}\mathrm{d}t$.

### 2.4.2. Adaptive Cuckoo Search (ACS) Algorithm

The CS algorithm's step ratio $R$ and discovery probability $P_a$ are both constants that are set. These two parameters can be adjusted to improve parameter optimization and performance of the CS algorithm.

(a) Improvements in step length ratios

The optimization process consists of two stages: global optimization and local optimization. A large step size is used in global optimization to improve convergence speed; a small step size is used in the local optimization stage to ensure convergence accuracy. However, a uniform step size setting cannot satisfy the above objectives at the same time, so the relevant parameters of the step size are adjusted to improve optimization performance of the CS algorithm.

In Equation (18), $R\frac{u}{|v|^{1/\beta}}\left[x_i^{(t)} - x_{\text{best}}^{(t)}\right]$ is set as the step size. It is considered that the step size is co-influenced by $R$, $u$, $v$, and $\left[x_i^{(t)} - x_{\text{best}}^{(t)}\right]$ while the degree of effect is uncertain. Therefore, in order to improve the global convergence speed and optimization accuracy of

the CS algorithm, $R$ is no longer set as a fixed parameter, and it is dynamically adjusted according to the following Equation (20):

$$R = R_s - \frac{k(R_s - R_e)}{G_{\max}}, \tag{20}$$

where $R_s$ and $R_e$ are the initial and final step size proportions, respectively, and $R_s > R_e$; $k$ is the current number of iterations; $G_{\max}$ is the maximum number of iterations.

(b)    Improvement in host detection probability.

The second rule of the algorithm is similar to the genetic algorithm chromosome mutation operation and is used to avoid the solution from falling into a local optimum. The value of discovery probability $P_a$ can have an impact on the performance of the algorithm. If the value of $P_a$ is too large, it will destroy the algorithm's search mechanism and even reduce it to a random search algorithm. If discovery rate $P_a$ is too small, the algorithm is prone to premature maturity.

Global optimization search focuses on searching all feasible domains. In the global optimization stage, a larger $P_a$ should be used for global variational search. To avoid the algorithm degenerating into a random search algorithm, $P_a$ should be appropriately reduced as the number of iterations increases. To summarize, $P_a$ is improved:

$$P_a = P_{ae}\left[1 + (P_{as}/P_{ae})^{1/(1+k)}\right], \tag{21}$$

where $P_{as}$ and $P_{ae}$ are the initial and final discovery rates, respectively.

### 2.4.3. Back Propagation Neural Network (BPNN)

Replacing the actual training process of the convolutional network with the network prediction process, the BPNN network should fulfill the following two requirements:(1) Simple structure and fast training speed; (2) Fast prediction speed and high accuracy. As a typical three-layer neural network, BPNN consists of only input, hidden and output layers, with a relatively simple structure [26,27], and has a strong prediction ability for nonlinear complex situations.

In this paper, we use a three-layer BPNN as a prediction model, assuming that the input hyperparameter combination vector is $x = (x_1, x_2, x_3)$. The outputs of its implicit and output layers can be expressed as, respectively:

$$\begin{cases} y_i = f\left(\sum_j w_{ij}x_j - \theta_j\right) \\ z_l = g\left(\sum_i h_{li}y_i - n_l\right) \end{cases}, \tag{22}$$

where $x_j$ is the $j$th feature; $w_{ij}$ and $h_{li}$ are the weight values of neuron connection; $\theta_j$ and $n_i$ are the threshold of the neuron.

The structure of the BPNN is schematically shown in Figure 3. In order for the BPNN to have the ability to predict the training accuracy of the convolutional network, the actual training accuracy of the corresponding convolutional network under different combinations of hyperparameters is counted so that a certain number of sample pairs are generated. The BPNN is allowed to train on these sample pairs to learn the nonlinear relationship between the values of the hyperparameters and the network training results, which is used as a prediction model to predict the fitness value of the optimization algorithm.

### 2.4.4. Agent Model of Adaptive Cuckoo Search Algorithm—BPNN

The design flow of the proposed agent model is shown in Figure 4. The reciprocal of the output result of the BPNN is used as the fitness value of the ACS: if the input value of the BPNN is ac, the fitness value is 1/ac. After the ACS algorithm reaches the maximum

number of iterations, the values of the training accuracy of the convolutional network and the combination of hyper-parameters that corresponds to the optimal algorithm can be output.

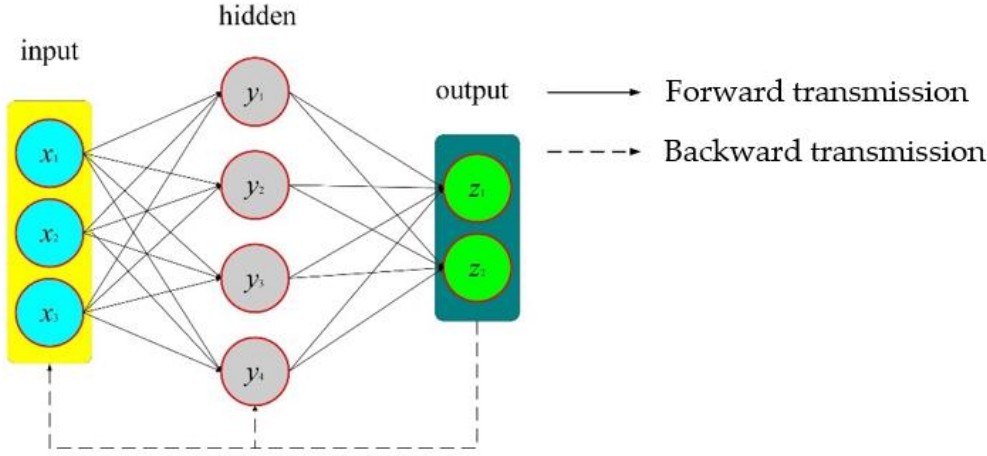

**Figure 3.** Schematic diagram of BPNN structure.

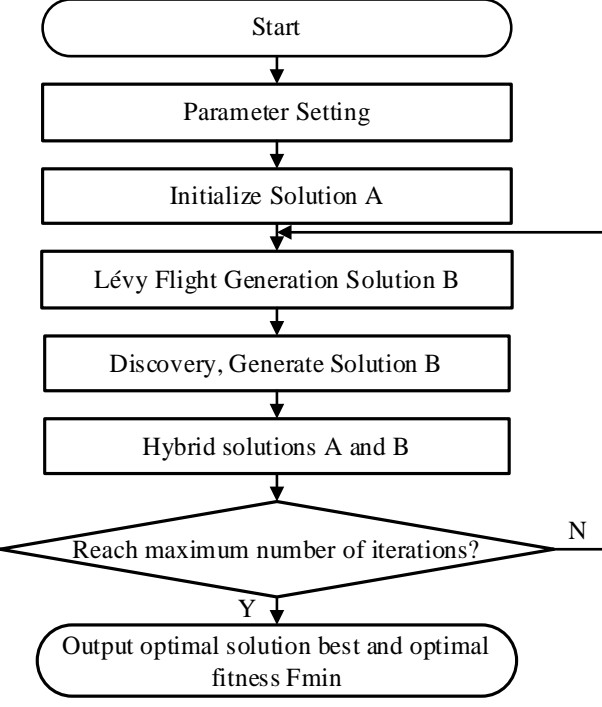

**Figure 4.** Flow chart of the adaptive cuckoo algorithm.

The compilation flow of the improved cuckoo algorithm is shown in Figure 4. Parameter initialization: $n$ is the solution space or the number of nests.

(a) Parameter setting. the number of nests, $n$, the maximum breeding algebra, $G_{max}$, the upper and lower boundaries of the solution, $Ub$ and $Lb$, the initial and final step sizes, $R_s$ and $R_e$, and the initial and final host discovery rates, $P_{as}$ and $P_{ae}$ are set.

(b) Initializing the solution. Solution A in each nest is initialized.

(c) Lévy flight. Lévy flight is implemented for Solution A to generate Solution B.

(d) Discovery. The host discovers partial Solution B according to probability $P_a$. For the discovered part, it is randomly generated again to form a new solution, B.

(e) Mix. The fitness values of Solution B and Solution A are compared, recording the medium and excellent Solution B in A, forming a new solution, still recording as A, and recording the current optimal solution best and optimal fitness $f_{\min}$.

(f) Determination of whether to terminate. If multiplication algebra reaches $G$, the optimal solution best and optimal fitness $f_{\min}$ are output, and the calculation is completed; otherwise, Step (c) is repeated and multiplication continues.

### 2.4.5. Analysis of Algorithm Improvement Effect

This section compares the performance of Particle Swarm Optimization (PSO), the Genetic Algorithm (GA), the CS algorithm and the ACS algorithm for optimization through simulation experiments. Performance test functions of the intelligent optimization algorithm are shown in Table 1. Figure 5 illustrates the 3D surface shapes of some of the test functions in Table 1. Obviously, the shapes of these test functions are characterized by complexity and nonlinearity, contain multiple peaks and valleys, and have a large number of local extreme points, which can be used to test the algorithm's global search performance, population diversity retention ability, algorithm resistance to premature maturity, local optimization search performance, and stability [28,29].

**Table 1.** Performance test function.

| Function Name | Function Expression | Parameter Value Range | Theoretical Optimal Solution (Minimum Value) |
|---|---|---|---|
| Goldstein and Price | $f_1(x) = \left\{ \begin{array}{l} \left[1 + (x_1 + x_2 + 1)^2 \left( \begin{array}{l} 19 - 14x_1 + 3x_1^2 \\ -14x_2 + 6x_1x_2 + 3x_2^2 \end{array} \right)\right] \\ \times \left[30 + (2x_1 - 3x_2)^2 \times \left( \begin{array}{l} 18 - 32x_1 + 12x_1^2 \\ +48x_2 - 36x_1x_2 + 27x_2^2 \end{array} \right)\right] \end{array} \right\}$ | $[-2, 2]$ | 3 |
| Branin | $f_2(x) = \left\{ \begin{array}{l} \left(x_2 - \frac{5}{4\pi^2}x_1^2 + \frac{5}{\pi}x_1 - 6\right)^2 \\ +10\left(1 - \frac{1}{8\pi}\right)\cos(x_1) + 10 \end{array} \right\}$ | $[-5, 15]$ | 0.397887 |
| Schaffer F6 | $f_3(x) = \frac{\sin^2\sqrt{x_1^2 + x_2^2} - 0.5}{\left[1 + 0.001\left(x_1^2 + x_2^2\right)\right]^2} - 0.5$ | $[-100, 100]$ | $-1$ |
| Rastrigin | $f_4(x) = \sum\limits_{i=1}^{D} \left[x_i^2 - 10\cos(2\pi x_i) + 10\right]$ | $[-5.12, 5.12]$ | 0 |
| Michalewicz | $f_5(x) = \sum\limits_{i=1}^{D} \sin(x_i)\left[\sin\left(\frac{ix_i^2}{\pi}\right)\right]^{2m}, m = 10$ | $[0, \pi]$ | $-4.6877$ (if D = 5) |
| Schwefel | $f_6(x) = 418.9829D - \sum\limits_{i=1}^{D} x_i \sin\sqrt{|x_i|}$ | $[-500, 500]$ | 0 |

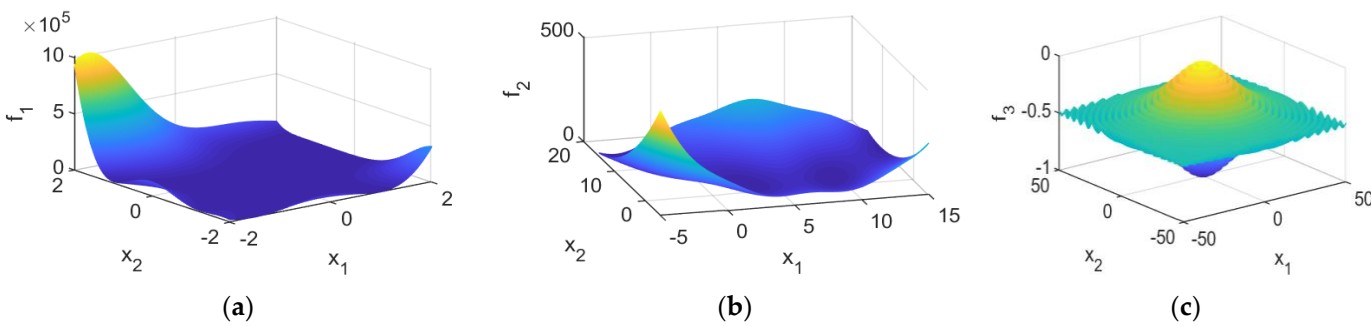

**Figure 5.** 3D shapes of partial test functions. (**a**) Goldstein and Price; (**b**) Branin; (**c**) Schaffer F6.

The performance of ACS is tested below. PSO, GA, CS and ICS are used to search for the minimum of the above test functions, respectively. All algorithms have population size $n = 30$ and maximum number of iterations $G_{\max} = 100$ generations. Each test function is searched 100 times each, and the final optimal function value of each test function is

recorded, $F_i$. Performance of each optimization algorithm is measured in terms of minimum error, maximum error and average error:

$$\text{Minimum error}: \text{BE} = \min(F_i) - F_{\text{to}}, \tag{23}$$

$$\text{Maximum error}: \text{WE} = \max(F_i) - F_{\text{to}}, \tag{24}$$

$$\text{Mean error}: \text{AE} = \text{mean}(F_i) - F_{\text{to}}. \tag{25}$$

The smaller the minimum error, the better the optimization algorithm's local optimization ability. The larger the maximum error, the worse the optimization algorithm's ability to resist premature maturity. The smaller the average error, the more stable the optimization algorithm's optimization performance. In short, the smaller the value of these three indicators, the better the performance of the optimization algorithm.

The test results are shown in Table 2. Comparing the test results of PSO, GA and CS, among these three evaluation indexes, except for the minimum error BE corresponding to test function $f_1$ and maximum error WE corresponding to test function $f_5$, the CS test indexes are the smallest, so the CS algorithm has better performance. Then, comparing the test results of CS and ACS, all the indexes of ACS test results are further reduced, so the performance of the ACS algorithm is better than that of CS. In conclusion, among these four algorithms, ACS has the best performance in finding the best value, and the multi-intelligence body strategy improves the performance of the CS algorithm.

**Table 2.** Performance test results of intelligent optimization algorithms.

| Algorithm Index | Test Function—Dimension | $f_1$—2 | $f_2$—2 | $f_3$—2 | $f_4$—5 | $f_5$—5 | $f_6$—5 |
|---|---|---|---|---|---|---|---|
| PSO | BE | $1.510 \times 10^{-6}$ | $5.275 \times 10^{-7}$ | $3.860 \times 10^{-3}$ | 0.007 | 1.512 | 398.4 |
| | WE | 0.301 | 2.308 | $4.595 \times 10^{-2}$ | 5.970 | 2.901 | 978.4 |
| | AE | $4.866 \times 10^{-2}$ | $2.311 \times 10^{-2}$ | $1.622 \times 10^{-2}$ | 2.127 | 2.286 | 700.7 |
| GA | BE | $6.490 \times 10^{-7}$ | $3.773 \times 10^{-7}$ | $1.072 \times 10^{-3}$ | 0.337 | 0.215 | 1.233 |
| | WE | 27.00 | $2.436 \times 10^{-2}$ | $3.725 \times 10^{-2}$ | 8.839 | 1.329 | 503.4 |
| | AE | 0.271 | $4.249 \times 10^{-4}$ | $1.127 \times 10^{-2}$ | 3.509 | 0.725 | 238.9 |
| CS | BE | $1.151 \times 10^{-6}$ | $3.686 \times 10^{-7}$ | $3.301 \times 10^{-5}$ | 0.007 | 0.199 | 1.121 |
| | WE | $3.957 \times 10^{-2}$ | $6.171 \times 10^{-4}$ | $9.716 \times 10^{-3}$ | 2.587 | 1.445 | 270.2 |
| | AE | $2.342 \times 10^{-3}$ | $1.880 \times 10^{-4}$ | $7.199 \times 10^{-3}$ | 0.895 | 0.685 | 59.21 |
| ACS | BE | $1.337 \times 10^{-8}$ | $2.295 \times 10^{-7}$ | 0 | 0 | 0.162 | 0.731 |
| | WE | 0.144 | $5.217 \times 10^{-4}$ | 0 | 0 | 1.029 | 125.7 |
| | AE | $1.759 \times 10^{-3}$ | $1.009 \times 10^{-4}$ | 0 | 0 | 0.356 | 21.15 |

## 3. Fault Diagnosis Algorithm Flow

So far, the framework of the small-sample transfer learning fault diagnosis method for the UAV engine based on the MSSST and the ACS-BPNN agent model optimized deep convolution is shown in Figure 6.

Specific processes are included:

(a) UAV engine fault presetting experiments are conducted to collect and preprocess the vibration signals in the fault state;

(b) The acquired vibration signals are converted into three-channel color time-frequency image samples by the MSSST and divided into the training set, the test set, and the validation set according to a certain ratio;

(c) A pre-trained ResNet-18 network model on the ImageNet image set is used as the base migration model;

(d) The learning rate of all network layers with parameter space before the last fully connected layer of the binary convolutional network is set to zero, i.e., these network layers are frozen and only the parameter-initialized connected layer of the last fully

connected layer is retained in order to learn the classification features of UAV engine fault samples;

(e) After training the network using the training set to obtain better training accuracy, the hyperparameters of the two types of convolutional networks are autonomously optimized using the ACS-BPNN agent model;

(f) Two types of convolutional networks are trained using optimized hyperparameters and the trained networks are used to classify the test samples for fault diagnosis.

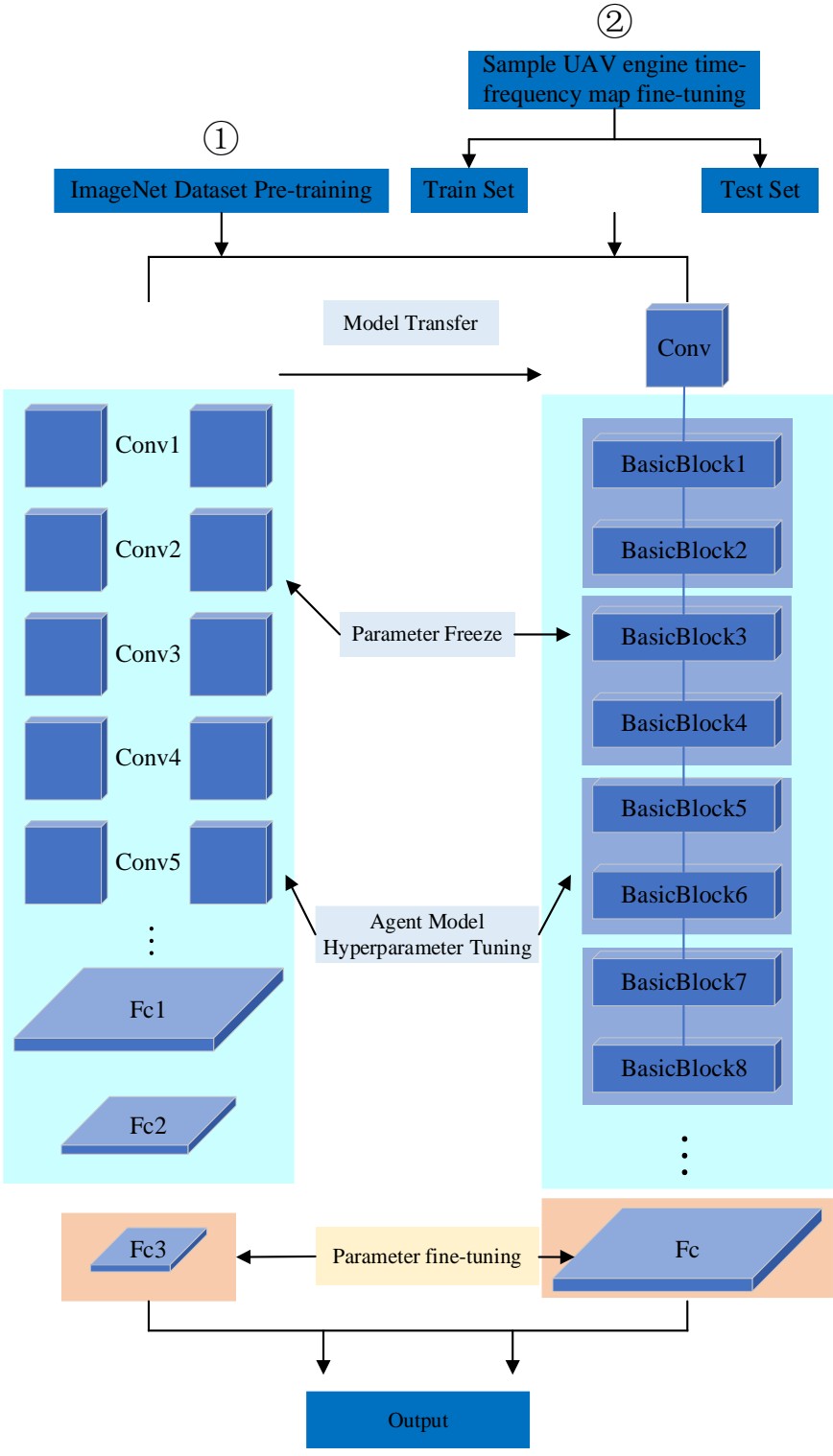

**Figure 6.** Flow chart of the fault diagnosis algorithm.

## 4. Experimental Data Collection and Description

The data acquisition experiment is carried out. The relevant technical parameters of the UAV engine used in the experiment are shown in Table 3, and the experimental data acquisition and related software analysis are shown in Figure 7. During the experiment, the vibration acquisition frequency is 20 k Hz, the length of experimental data acquisition is 120 s, and the vibration equipment used is the DHDAS dynamic signal acquisition and analysis system.

**Table 3.** Basic technical parameters of the UAV engine.

| Type | Parameter | Category | Parameter |
|---|---|---|---|
| Bore | 79.5 mm | Weight | 75.0 kg |
| Piston stroke | 60 mm | Maximum continuous speed | 5500 r/min |
| Number of cylinders | 4 | MCR | 1250 hPa |

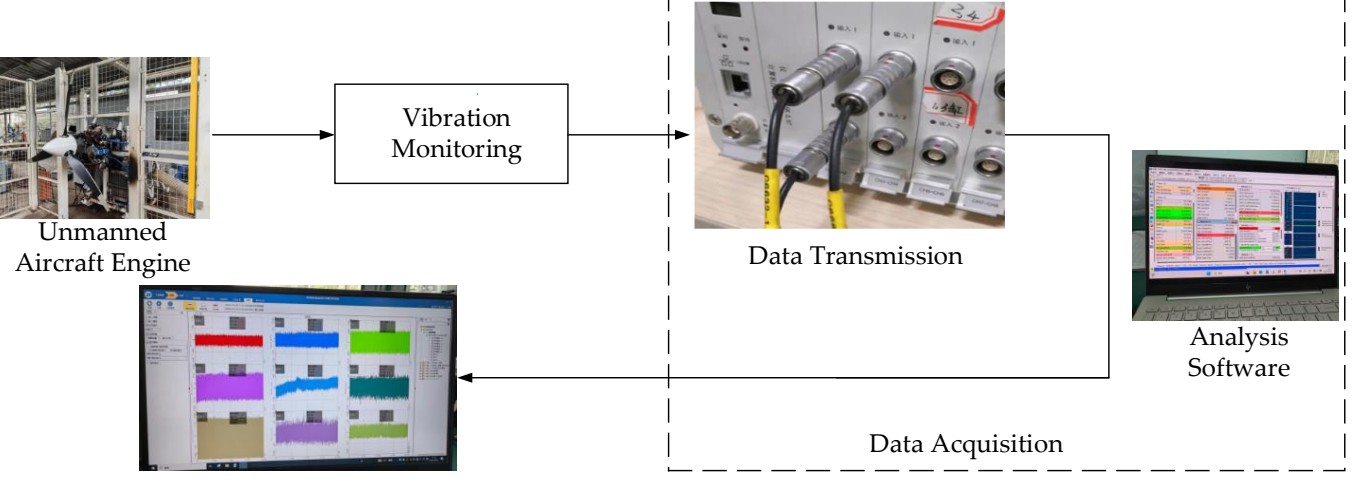

**Figure 7.** Data acquisition and analysis process.

In the experiment, the model of the vibration sensor and its installation position are shown in Figure 8.

According to the frequent faults of UAV engines, this paper presets four different fault states. The preset fault experiment is shown in Figure 9. The specific descriptions of the four different fault states are as follows:

(a) Simulation of the four-cylinder engine does not inject fuel (injector failure).

The engine four-cylinder injector plug is removed, the four-cylinder does not work, the engine is running in the failure mode of the lack of cylinders. Engine working conditions are as follows: speed 3000 r/min, throttle 11%.

(b) Simulation of the four-cylinder engine does not work (spark plug connector abnormal).

The engine four-cylinder high-pressure shielding cap is removed, four-cylinder oil supply is normal but cannot be ignited, the four-cylinder cannot work properly, the engine is still in the lack of cylinder fault mode operation. Engine working conditions are as follows: speed 3000 r/min, throttle 11%.

(c) Low voltage on the analog engine supply.

The normal supply voltage of the EFI engine is 13~14V, and the adjusted supply voltage is only 8V, the engine is running in the low-voltage fault mode, and the engine working conditions are as follows: speed 3000 r/min, throttle 11%.

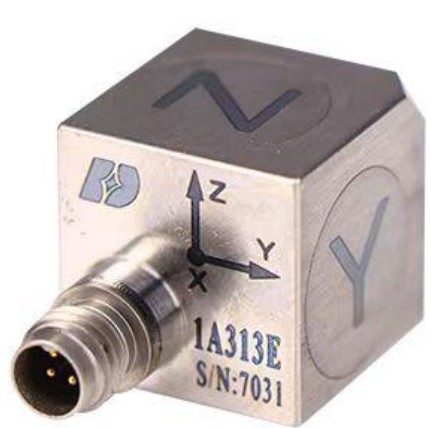

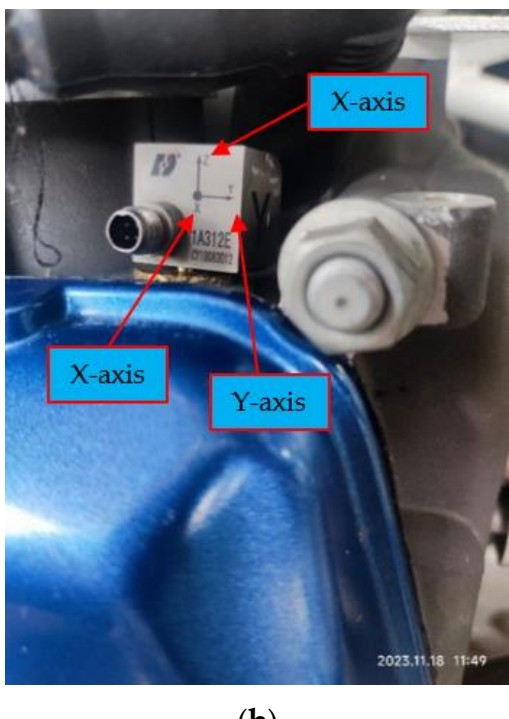

(**a**)          (**b**)

**Figure 8.** Sensor and its installation position. (**a**) The shape of the sensor; (**b**) The installation position of the sensor.

(d)   Simulation of disconnecting the A ignition (disconnecting one-way ignition).

The engine is in a normal condition with A and B ignitors, with the A ignitor controlling the upper ignition system of the four cylinders and the B ignitor controlling the lower ignition system of the four cylinders. After the A igniter is disconnected, the engine is running in the failure mode of the upper ignition system of the four cylinders, and the engine working conditions are as follows: speed 3000 r/min, throttle 11%.

(e)   The engine is in a normal working condition.

The working conditions are the following: speed 3000 r/min, throttle 11%.

In the above preset fault experiments, it is shown that the experimental process is in accordance with the vibration acquisition frequency of 20 k Hz. the vibration equipment used is that of the DHDAS Dynamic Signal Acquisition and Analysis System for the collection of experimental data and the initial software display; however, it is only obtained through the collection of the data software and not from the time-domain diagram of the experimental equipment. Therefore, we are unable to determine the failure of the situation, and there is a need to use a further processing method to achieve the diagnosis of the fault.

The fault diagnosis method of the MSSST with small-sample migration learning proposed in this paper is utilized to identify the individual states of UAV engines. As there are few studies on UAV engine fault diagnosis at present, there is often the problem of insufficient sample size. Therefore, 50 samples are randomly selected from each fault state during the experimental process of this paper, which are used to realize the fault diagnosis under small-sample conditions. Dividing the training set, the validation set and the test set according to the ratio of 7:2:1, 35 training samples, 10 validation samples and 5 test samples are obtained for feasibility validation experiments of the MSSST and small-sample migratory learning fault state recognition methods.

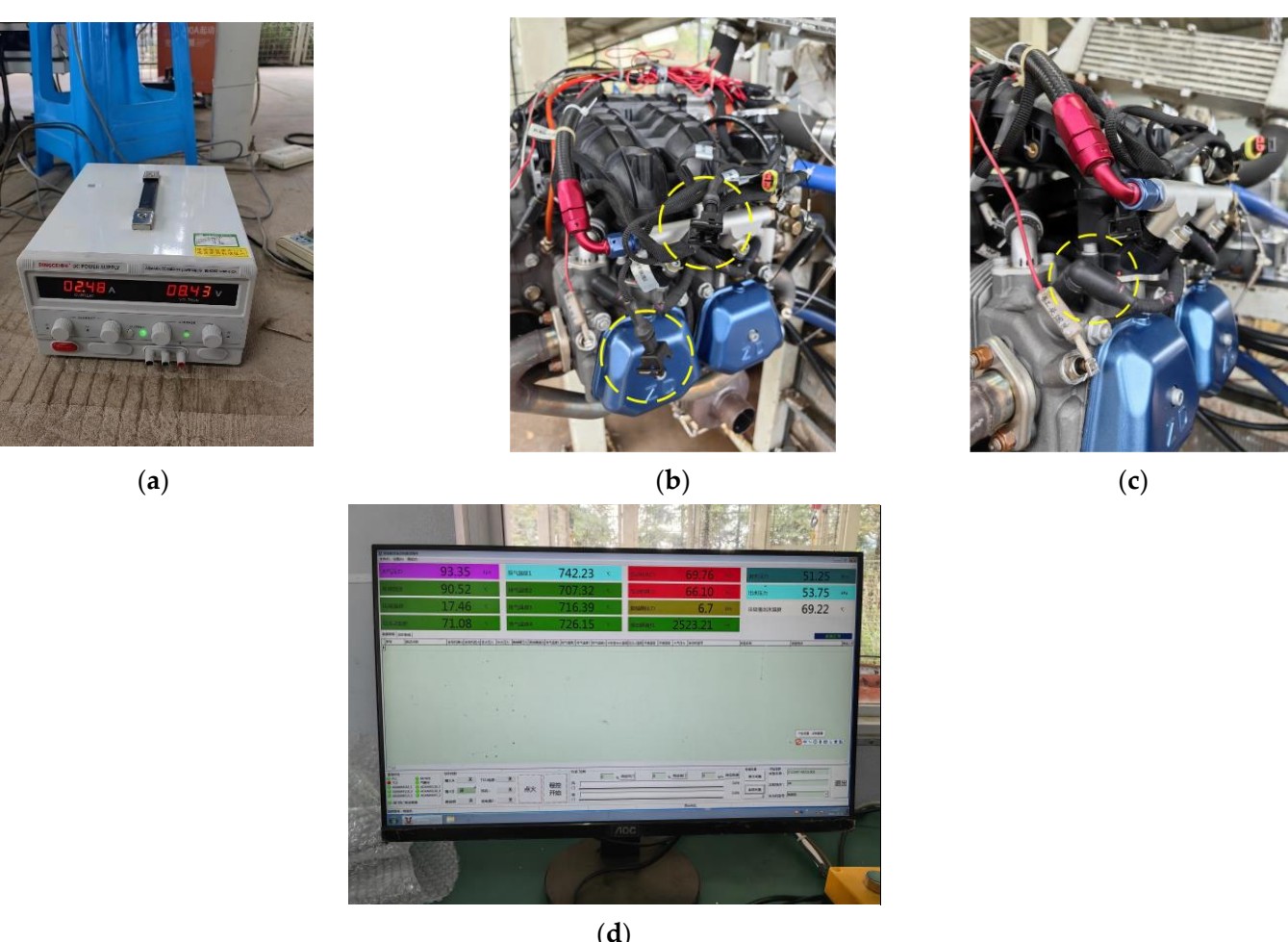

**Figure 9.** Preset Fault Experiment. (**a**) Insufficient power supply; (**b**) Fuel injector plug fault; (**c**) Spark plug abnormality; (**d**) Disconnecting 1 igniter.

The original vibration signal is processed by the MSSST to obtain the time-frequency diagram, as shown in Figure 10. The coordinate system, legend and blank parts are removed to avoid the impact on the accuracy of fault diagnosis.

From the figure, it can be seen that the time-frequency resolution accuracy of the time-frequency image obtained by the MSSST is very high, avoiding the phenomenon of time-frequency blurring. Analysis from this perspective is beneficial to the identification of subsequent samples. However, observing the time-frequency images in different states, it can be found that the differentiation between the time-frequency images is not high, and the fault classification of the UAV engine cannot be realized only by time-frequency images.

In order to compare the effectiveness of the methods proposed in this paper, experiments are conducted to compare the training accuracy of different models. During the experimental comparison, four different comparison methods are set:

(a)    SSST-CS-BPNN-ResNet-18;
(b)    MSSST-CS-BPNN-ResNet-18;
(c)    SSST-ACS-BPNN-ResNet-18;
(d)    MSSST-ACS-BPNN-ResNet-18.

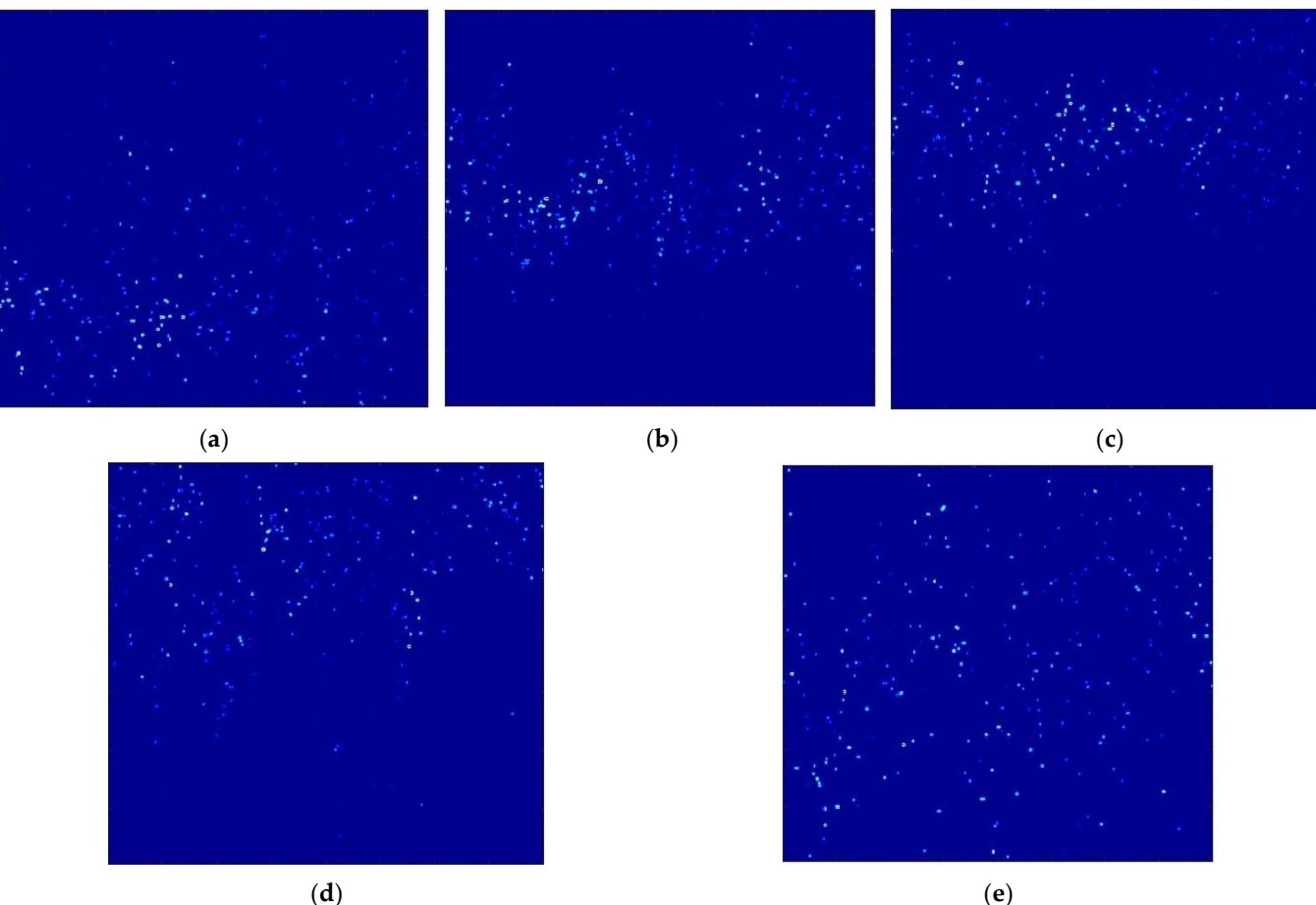

**Figure 10.** Time-frequency images of vibration signals in different statuses. (**a**) Status 1; (**b**) Status 2; (**c**) Status 3; (**d**) Status 4; (**e**) Status 5.

Analyzing the results shown in Figure 11 and Table 4 and comparing the results of Figure 11a,b, as well as Figure 11c,d, it can be seen that the MSSST method proposed in this paper can improve the accuracy of fault diagnosis to a certain extent, indicating that the time-frequency diagram obtained using MSSST analysis is superior to those obtained using traditional time-frequency analysis methods such as SSST. Comparing Figure 11a,c, and Figure 11b,d, it can be found that the performance of the ResNet-18 network optimized by ACS-BPNN is improved, and the accuracy of fault diagnosis is also improved. In general, the proposed method is optimized in time-frequency resolution and fault diagnosis results.

In order to show more clearly the fault classification process of the method proposed in this paper, the said method is visualized in three dimensions, and the fault classification is obtained as shown in Figure 12. From Figure 12, it can be clearly seen that the method can realize the classification of five different states and carry out the fault diagnosis of a UAV engine under small-sample conditions.

**Table 4.** Comparative results between different methods.

| Methodologies | Accuracy |
| --- | --- |
| SSST-CS-BPNN-ResNet-18 | 93.5028 |
| MSSST-CS-BPNN-ResNet-18 | 95.3642 |
| SSST-ACS-BPNN-ResNet-18 | 96.0625 |
| MSSST-ACS-BPNN-ResNet-18 | 97.1751 |

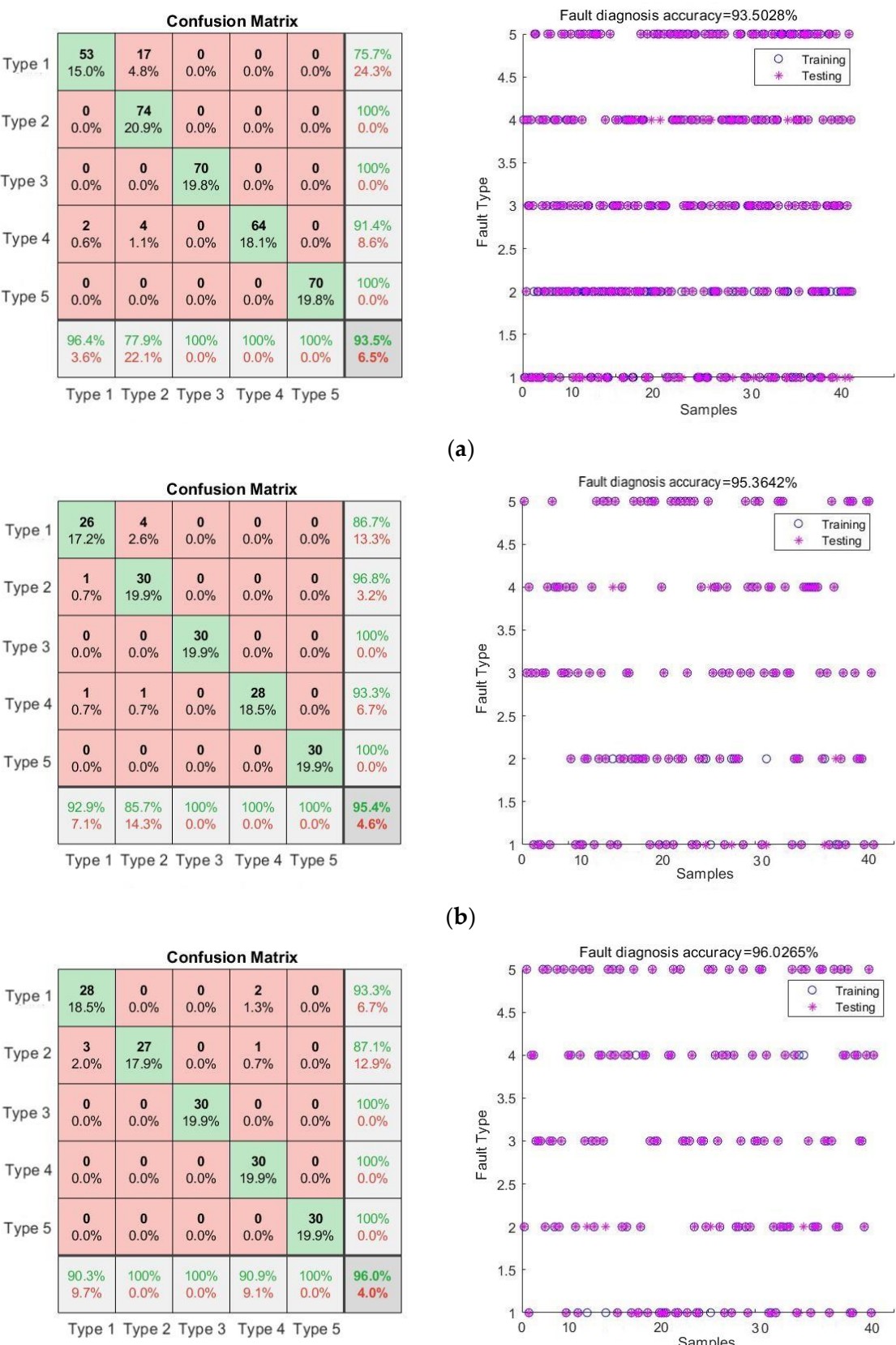

**Figure 11.** *Cont.*

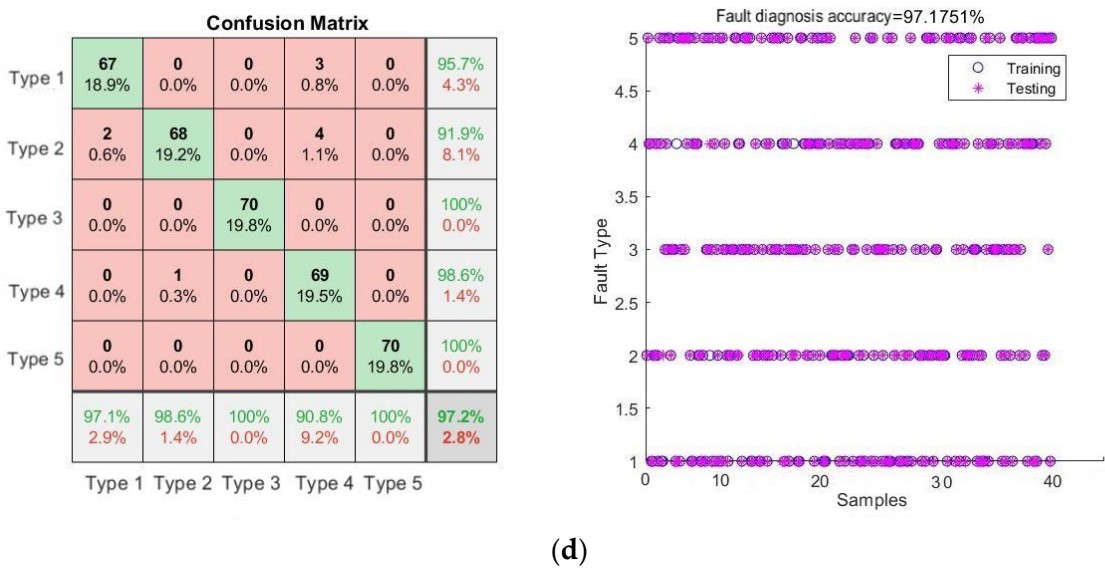

**Figure 11.** Comparative images between different methods. (**a**) SSST-CS-BPNN-ResNet-18; (**b**) MSSST-CS-BPNN-ResNet-18; (**c**) SSST-ACS-BPNN-ResNet-18; (**d**) MSSST-ACS-BPNN-ResNet-18.

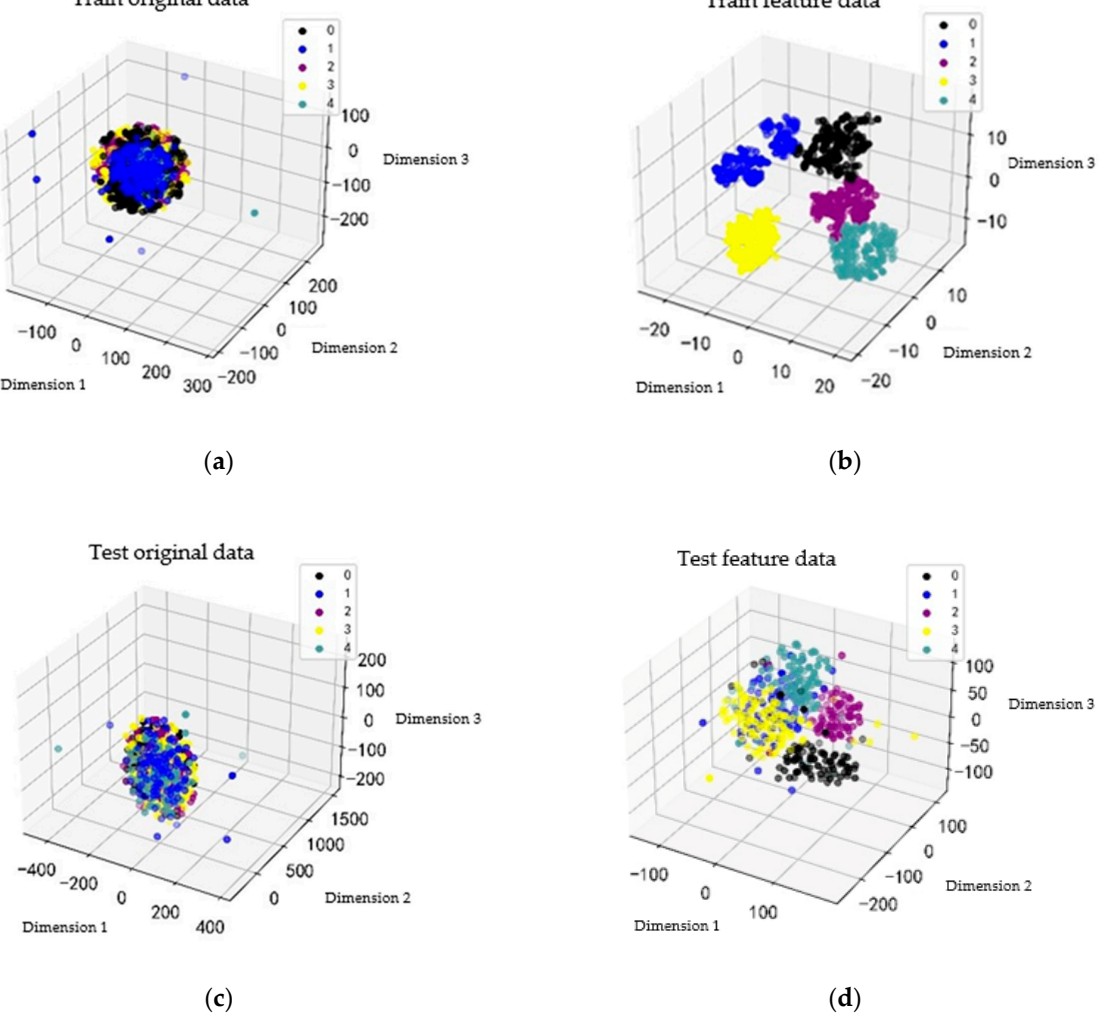

**Figure 12.** The fault classification results of the proposed method. (**a**) Training set raw data; (**b**) Validation set raw data; (**c**) Training set feature data; (**d**) Validation set feature data.

## 5. Conclusions

In this paper, we study the small-sample transfer learning fault diagnosis method for a UAV engine based on the MSSST and ACS-BPNN agent model optimized deep convolution. The MSSST time-frequency image of the vibration signal of the UAV engine is used as a sample, and the hyper-parameters of the ResNet-18 network are autonomously optimized using the ACS-BPNN agent model. The main conclusions of the proposed method in this paper are shown below.

(a) The vibration signals of the UAV engine under different fault statuses are collected by vibration sensors, and the vibration signals are converted into time-frequency diagrams using MSSST as inputs to the fault diagnosis model, and this feature extraction method minimizes the disturbing factors of human-selected features.

(b) The CS algorithm, i.e., the ACS algorithm, is improved, which effectively enhances the intelligence of the algorithm and further improves its global optimization capability. The combination of the ACS algorithm and the BPNN model is used for the hyperparameter autonomous optimization of convolutional network ResNet-18. It is experimentally verified that the proposed method can effectively diagnose the faults of UAV engines under small-sample conditions.

In summary, the existing fault diagnosis algorithms usually require sufficient data samples compared with the algorithm proposed in this paper, which requires a small amount of data. Therefore, this study can provide theoretical reference and technical support for fault diagnosis in the case of small samples. However, there are still some problems and deficiencies in this study, such as the need to optimize the deep network structure, research more accurate agent models, find more effective migration strategies to reduce the data volume requirement, etc. Our team agrees that these problems are the next focus of the research direction.

**Author Contributions:** Conceptualization, conceptualization, S.L. and Z.L.; methodology, Z.L.; software, Y.H. and S.L.; validation, Z.L., S.L. and Y.Y.; formal analysis, K.H. and Y.Y.; investigation, S.L. and S.M.; resources, Z.C. and S.L.; data curation, K.H.; writing—original draft preparation, S.L.; writing—review and editing, S.L.; visualization, X.M. and S.L.; supervision, Z.C.; project administration, Z.C.; funding acquisition, Y.Y and K.H. All authors have read and agreed to the published version of the manuscript.

**Funding:** This research was funded by National Defense Research Fund Project (Grant No. 212LJ44004).

**Data Availability Statement:** The authors confirm that the data supporting the findings of this study are available within the article.

**Conflicts of Interest:** The authors declare no conflicts of interest.

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
