# Peer review of "Research on a Small-Sample Fault Diagnosis Method for UAV Engines Based on an MSSST and ACS-BPNN Optimized Deep Convolutional Network"

_processes, doi:10.3390/pr12020367_

Round 1

Reviewer 1 Report

Comments and Suggestions for Authors

Comments and suggestions

Ms. No.: processes-2830036

Title: Research on small samples fault diagnosis method for UAV engine based
on MSSST and ACS-BPNN optimized deep convolutional network

Aiming at the difficulty of extracting fault information in the faulty status of UAV engine, and the high time cost and large data requirement of the existing deep learning fault diagnosis algorithms with many training parameters. In this paper, small samples transfer learning fault diagnosis algorithm is proposed. Firstly, the vibration signals under the engine fault status are converted into a two-dimensional time-frequency map by multiple simultaneous squeezing S-transform (MSSST), which reduces the randomness of manually extracted features. Secondly, to address the problems of slow network model training and large data samples requirement, a transfer diagnosis strategy using the fine-tuned time-frequency map samples as the pre-training model of ResNet-18 convolutional neural network is proposed. In addition, in order to improve the training effect of the network model, an agent model is introduced to optimize the hyperparameter network autonomously. Finally, experiments show that the algorithm proposed in this paper can obtain high classification accuracy in the fault diagnosis of UAV engine compared with other commonly used methods and maintains a very stable small-sample transfer learning capability under this condition.

Some comments about this paper:

1)   In Table 2, the efficiency of the examined algorithms is almost identical; nevertheless, the ACS performs better with smaller values in the high order of the error. What was the rationale behind the authors' consideration of ACS as the superior algorithm?

2)   In Table 2, which criterion is deemed more important and given priority?

3)   In the introduction, consider exploring various machine learning scenarios and optimization methods. The papers below provide useful clarification on these aspects, making them valuable for recommendation.

10.3390/a14040119

10.1038/s41598-023-47174-w

10.1016/j.isatra.2021.07.043

4)   What criteria were used to choose the parameters and ranges outlined in Table 3?

5)   What led to the selection of the 7:2:1 approach?

6)   Figure 11; The fault diagnosis accuracy among the four methods is comparable. The distinctions between methods (b) and (d) compared to (a) and (c) are not significantly pronounced.

7)   What is the intended purpose of Figure 12?

8)   Provide a detailed physical discussion of the results obtained.

Concluding, (major revision).

Reviewer 2 Report

Comments and Suggestions for Authors

In the abstract, include obtaining accuracy % and discussing other methods you compared.

In the abstract, expand UAV.

The scope of the paper needs to be highlighted in the introduction.

A comparative table at the end of the related work section is needed to illustrate ‎the difference between existing works.

Figure 2,6 is not precise.

In discussion, table 2 explains with a value that is the best performance algorithm.

The dataset source needs to be explained in section 4.

In the abstract, the authors discussed that the proposed methods show better accuracy compared to existing methods. Authors must show the values in the table and graphically in section 4.

Suggest to authors, using the confusion matrix, to find other possible metrics shown with tables and graphs.

In conclusion, discuss the obtained results and explain.

I suggest a few more related papers to cite and refer to. This will enrich the survey.

S. Sundarraj, R. Vijaya Kumar Reddy, B. Mahesh Babu, G. H. Lokesh, F. Flammini, and Rajesh Natarajan, “Route Planning for an Autonomous Robotic Vehicle Employing a Weight-Controlled Particle Swarm-Optimized Dijkstra Algorithm,” IEEE Access, pp. 1–1, 2023, doi: 10.1109/ACCESS.2023.3302698.

Singh, P. (2023). Enhancing Performance of Hybrid Electric Vehicle using Optimized Energy Management Methodology. International Journal of Data Informatics and Intelligent Computing, 2(3), 1–10. https://doi.org/10.59461/ijdiic.v2i3.74

Comments on the Quality of English Language

Minor editing of the English language required

Round 2

Reviewer 1 Report

Comments and Suggestions for Authors

Accept

Reviewer 2 Report

Comments and Suggestions for Authors

Authors addressed/updated suggested comments.